# Simulation of Magnetic Resonance guided Laser Interstitial Thermal Therapy Temperature Maps through Time-series based Deep Learning Methods: Usage of ConvLSTM

**Saba Sadatamin**[1,2]                                    SABA.SADATAMIN@MAIL.UTORONTO.CA
**Paola Driza**[1]                                          PAOLA.DRIZA@MAIL.UTORONTO.CA
**Gemma Postill**[1]                                        GEMMA.POSTILL@UTORONTO.CA
**Steven Robbins**[3]                                       SROBBINS@MONTERIS.COM
**Richard Tyc**[3]                                          RTYC@MONTERIS.COM
**Rahul G. Krishnan**[1]                                    RAHULGK@CS.TORONTO.EDU
**Lueder A. Kahrs**[1]                                      LUEDER.KAHRS@UTORONTO.CA
**Adam C. Waspe**[1,2]                                      ADAM.WASPE@SICKKIDS.CA
**James M. Drake**[1,2]                                     JAMES.DRAKE@SICKKIDS.CA

[1] *University of Toronto, Toronto, Canada*

[2] *The Hospital for Sick Children, Toronto, ON, Canada*

[3] *Monteris Medical, Winnipeg, Manitoba, Canada*

**Editors:** Under Review for MIDL 2024

## Abstract

Magnetic resonance-guided laser interstitial thermal therapy (MRgLITT) is a minimally invasive therapy that leverages thermal ablation to treat drug-resistant focal epilepsy. Patient-specific heat sinks, such as blood vessels, complicate planning of MRgLITT as it creates patient-level variability in how heat from the laser propagates, potentially undermining treatment efficacy. We developed a deep learning framework to predict the resulting spatio-temporal temperature maps of the laser monitoring system. We used a convolutional long short term memory (ConvLSTM) network and evaluated performance using quantitative and impact evaluation metrics. In impact evaluation, we binarized temperature images with pixels exceeding a temperature of 39°C indicative of cell death. We then use a dice score and sensitivity metrics to evaluate the overlap between predicted and ground truth thermal dose margin. We demonstrate strong performance of our ConvLSTM framework (structural similarity index metric: 0.88, dice score: 0.85, and sensitivity: 0.77), demonstrating that predicted heat propagation was similar to ground truth. Our findings can be used by neurosurgeons to improve delivery of MRgLITT.

**Keywords:** magnetic resonance-guided laser interstitial thermal therapy (MRgLITT), deep learning, convolutional long short term memory (ConvLSTM), treatment monitoring

## 1. Introduction

Magnetic resonance-guided laser interstitial thermal therapy (MRgLITT) is a novel, minimally invasive thermal therapy for brain tumors and drug-resistant epilepsy, including mesial temporal lobe epilepsy (Chen et al., 2021; Lee et al., 2019; Medvid et al., 2015). The primary challenge with MRgLITT is predicting optimal thermal dose, which minimizes damage to healthy brain tissue while ensuring adequate lesion ablation and positive therapeutic outcomes (Gao et al., 2024). Such task is particularly complex when the treatment

area is surrounded by heat sinks, such as blood vessels and ventricles, which affect how heat is distributed. Addressing this challenge requires accurate preoperative computation of heat propagation in patient-specific environments. Previously, heat propagation was modeled using Finite Element Methods (FEM), but these methods lack access to dynamic optical properties, resulting in rough predictions (Lad et al., 2023). Our objective was to simulate laser heat propagation monitoring system relative to the specified mesial temporal lobe epilepsy (MTLE) lesion using a convolutional long short term memory (ConvLSTM) network. This work provides foundational knowledge needed to develop a simulation system for pre-operative planning to assist neurosurgeons in identifying the optimal laser location considering the dynamic tissue parameters using thermometry images.

## 2. Materials and Methods

Our approach leveraged ConvLSTM and the effect of time-dependencies between 2D frames (e.g., heat propagation representations) to predict the MRgLITT monitoring system (Figure 1). The study cohort was derived from a new private dataset comprising of 81 patients with drug-resistant MTLE receiving MRgLITT. Data captured includes sequences of 2D temperature maps called magnetic resonance thermometry (MRT), which are produced by the monitoring system and reconstructed from the MRI phase image and associated temperature change(Rieke and K, 2008). Across the 81 patients, the dataset included 319 acquisitions each formatted as DICOM images. Each acquisition reflects the release of laser energy and subsequent generation of MRT images. The images were cropped from the center to 51x51 pixels treatment area to encode the laser location and avoid learning sources of noise during model training.

### 2.1. Model Architecture and Training

We used ConvLSTM, a recurrent neural network, (Balderas et al., 2019) to leverage spatio-temporal features of sequential MRT acquisitions. The model was trained on consecutive sequences of MRTs (using 51x51 positional encoded inputs, centred by the laser location) from timestamps $t_0$ to $t$. We predicted the subsequent 51x51 MRT at $t + 1$. We trained ConvLSTM network multiple times for different timestamps $t$ ($t = 4, 9, 14, 19, 24$) and reported average performance for the prediction of $t + 1$ timestamp.

### 2.2. Evaluation

Quantitatively, we evaluated pixel temperature predictions from the model using two metrics: root mean square error (RMSE) and structural similarity index (SSIM), both previously used to evaluate 2D snapshot MRT predictions using U-Net (Sadatamin et al., 2024; Ronneberger et al., 2015). To report MRgLITT impact, we binarized ground truth and predicted MRT images and considered pixel values of more than 39 as ablated tissue following the CEM43 metric (Pearce, 2013; MacLellan et al., 2018). The CEM43 metric represents the cumulative exposure of tissues to elevated temperatures (Pearce, 2013). After binarization, we evaluated the model with dice scores and classification sensitivity. We used dice score to quantify the overlap of ablated tissue between predicted and ground truth MRTs and classification sensitivity to report the number of pixels correctly classified as ablated

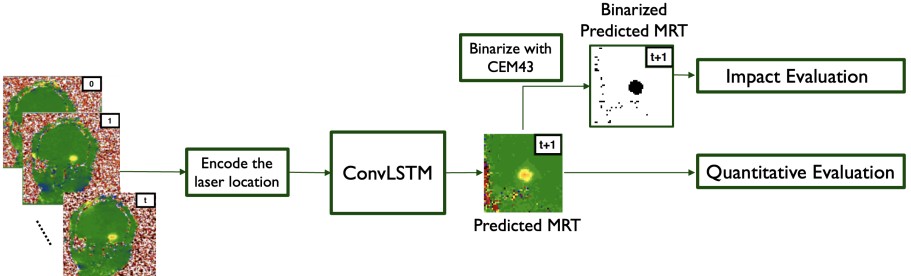

Figure 1: ConvLSTM Framework - We encoded the position and then fit the stacked images as a sequence to the model to predict the $t + 1^{th}$ MRTs from ConvLSTM.

tissue (vs. non-ablated tissue). Both metrics range from 0 to 1 inclusively, with 0 indicating no overlap and 1 indicating perfect overlap between predicted and ground truth images.

## 3. Results and Discussion

The visualization of ConvLSTM output is displayed in Figure 2. The average performance of of our framework across all $t+1^{th}$ predictions, as measured by the RMSE, reveals a pixel-to-pixel temperature difference between its predictions and the ground truth of approximately $3°C$ within the treatment area. The SSIM scoring 0.88, indicates high overall model similarity. Post binarization, the model demonstrated a dice score of 0.85 and a sensitivity of 0.77 for correctly classifying ablated tissue. These predictions are promising not only when measured against various evaluation metrics but also as they remain within the $5°C$ variatiom, a clinically accepted threshold for neurosurgeons. This is commendable in light of the $1°C$ degree of uncertainty already present in the monitoring system (de Senneville et al., 2007). In conclusion, we employed a ConvLSTM framework to anticipate the distribution of heat propagation in brain tissue in real-time, using the initial t time intervals to predict the $t + 1^{th}$.

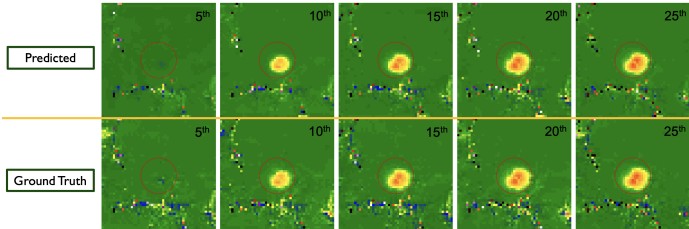

Figure 2: Output of the ConvLSTM frameworks to predcit the $t + 1^{th}$ MRT generated by the ConvLSTM of the MRgLITT monitoring system. The red circle indicates the treatment area that we used in our evaluations.

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
