# OpenReview forum: "Simulation of Magnetic Resonance guided Laser Interstitial Thermal Therapy Temperature Maps through Time-series based Deep Learning Methods: Usage of ConvLSTM"
_MIDL.io/2024/Short_Papers — MIDL 2024 Short Papers_

### Official Review · Reviewer_m3gi · 2024-04-24

**Confidence:** 4
**Final Rating:** 3.5

**Review:**

This paper introduces a novel application of deep learning for Magnetic resonance-guided laser interstitial thermal therapy (MRgLITT), a minimally invasive treatment for epilepsy. The authors develop a deep learning framework using a Convolutional Long Short-Term Memory (ConvLSTM) network to analyze sequences of temperature maps acquired during MRgLITT, predicting future temperature distributions and effectively identifying ablated tissue based on temperature thresholds. The model achieves high accuracy, with an average pixel-to-pixel difference of only 3°C compared to actual measurements.

Limitations of the study include reliance on Magnetic Resonance Thermometry (MRT) for ground truth temperature data, limited generalizability as the study focuses on mesial temporal lobe epilepsy (MTLE), and a lack of discussion on computational efficiency or latency for real-time implementation.
Comparison to state-of-the-art (SOTA) methods is not explicitly addressed in the paper. However, potential comparisons could be made with physics-based simulations for heat propagation prediction and alternative deep learning models like U-Net for medical image segmentation tasks.

---

### Decision · Program_Chairs · 2024-04-26

Accept